# A Systematic Review of Dynamic Forces and Kinematic Indicators of Front and Roundhouse Kicks across Varied Conditions and Participant Experience

**DOI:** 10.3390/sports11080141

**Published:** 2023-07-28

**Authors:** Michal Vagner, Daniel John Cleather, Vladan Olah, Jan Vacek, Petr Stastny

**Affiliations:** 1Department of Sports Games, Faculty of Physical Education and Sport, Charles University in Prague, 162 52 Prague, Czech Republic; danceleather@gmail.com (D.J.C.); j.vacek.mail@gmail.com (J.V.); stastny@ftvs.cuni.cz (P.S.); 2Department of Military, Faculty of Physical Education and Sport, Charles University in Prague, 162 52 Prague, Czech Republic; olah@ftvs.cuni.cz; 3Faculty of Sport, Health and Applied Science, St. Mary’s University, Waldegrave Road, Twickenham TW1 4SX, UK

**Keywords:** biomechanics, martial arts, impact force, maximum velocity

## Abstract

Impact force and maximum velocity are important indicators of kick efficiency. Therefore, this systematic review compared the front kick (FK) and roundhouse kick (RK), including their impact force, maximum velocity, angular velocity, and execution time, considering various target types and experience levels. Following PRISMA guidelines, the Web of Science, SportDiscus, and PubMed were systematically searched for articles published from January 1982 to May 2022. Normalized kicking values were compared using one-way ANOVA. Eighteen articles included FKs (sample: 113 elite men, 109 sub-elite men, and 46 novices), and twenty-five articles included RKs (sample: 238 elite men, 143 sub-elite men, and 27 novice men). The results indicate that the impact force of the FK were 47% (*p* < 0.01), 92% (*p* < 0.01), and 120% (*p* < 0.01) higher than those of the RK across novice, sub-elite, and elite groups, respectively. Moreover, the maximum foot velocity of the RK was 44% (*p* < 0.01) and 48% (*p* < 0.01) higher than that of the FK for the sub-elite and elite groups, respectively. Furthermore, the elite group had 65% (*p* < 0.01) higher knee extension angular velocity with the RK than with the FK and 138% (*p* < 0.01) higher hip extension angular velocity with the FK than with the RK. In summary, the findings suggest that the FK is more effective in generating forceful kicks, while the RK has the potential for rapid execution.

## 1. Introduction

In the realm of fighting activities, such as martial arts, combat sports, and close combat, participants dedicate themselves to enhancing their strength and speed and refining their techniques to overcome opponents or break solid targets [1,2]. Biomechanical tools have proven instrumental in investigating and improving the techniques employed in these sports and combat activities, with researchers utilizing dynamic and kinematic indicators for thorough analyses [3,4]. However, as the number of studies continues to rise, researchers must consider the varying conditions under which combat techniques are performed. Consequently, comparing findings from previously published studies to establish defined qualities of kicking techniques is essential.

While previous studies have explored various aspects of kicks, a comprehensive comparison of key characteristics between the front kick (FK) and roundhouse kick (RK) is lacking. This systematic review aims to bridge this gap by comparing the impact forces, maximum velocities, and execution times of the FK and RK across different target distances, target types, and experience levels. By examining these performance attributes, this study seeks to provide practical insights into the differences between and practical applications of the FK and RK, shedding light on the dynamic and kinematic variations at different technical levels and under different execution conditions.

The kick is among the fundamental techniques employed in combat activities to overcome opponents, which requires maximum strength and speed.

Previous research has often compared kicks to punches and explored comparisons among different experience levels, genders, and fighting actions [5,6,7,8,9,10,11]. Notably, studies have examined kicks at various distances from the target, with varying types of targets, and even under various carried military loads [12,13,14,15,16,17]. However, a notable gap exists in the comparison of the key characteristics between different types of kicks, explicitly focusing on the frequently reported FK and RK.

Significant differences exist in the execution of the FK and RK, particularly in the lower limb track and swing phases involving hip and knee coupling, particularly in terms of hip flexion and knee extension [18]. The FK entails a direct foot strike achieved through hip flexion and knee extension toward the opponent or target. Conversely, the RK involves a swinging arc of the hip and rapid knee extension to strike the target with the shin or instep. The technical execution of both kicks significantly influences accuracy, precision, and the ability to hit the target effectively [11,19], thereby relating to the performer’s level of proficiency. Consequently, the performance level can be a fundamental differentiating factor for the FK’s and RK’s net force and kicking speed characteristics.

Net force and kicking speed characteristics play vital roles in overcoming or neutralizing an opponent’s attack and are integral to determining the transferred momentum of energy [13,17,20,21,22,23,24]. These metrics offer valuable insights into the force production of a kick and contribute to a better understanding of its mechanics and effectiveness in combat sports and martial arts. The kicking speed, crucial for its efficacy, relies on factors such as execution time, which can often exceed the opponent’s reaction, as well as the reaction time itself [7,13,16,23]. Other variables include velocity, acceleration, and the angular velocity of the hip and knee [7,16,25,26,27,28,29,30].

A powerful technique, such as a kick, depends, in part, on the coordinated momentum of multiple body parts [11,22]. The front kick and roundhouse kick exemplify proximo-distal movements, where motion originates from joints and segments closer to the body’s center and progressively extends toward the extremities [31]. For instance, to achieve the maximum foot velocity in executing a front kick, athletes must increase the velocity of the knee as it travels toward the target [32]. On the other hand, for the optimal execution of a straight kick, the axial over-rotation of the hips in the sagittal plane becomes necessary to generate a higher impact force. Regarding the roundhouse kick, the velocity of the kicking foot results from the combined effect of linear motion at the pivot hip and the angular motion of the pelvis around the pivot hip, with the hip displacement contributing significantly in the initial phases [25]. However, it was found that circular kicks are executed in less time and at a higher foot velocity than other types of kicks but that, in the sagittal plane, they exhibit similar kinematic data (temporal parameters) to other kicks [14,25,33]. Moreover, differences in the execution technique, particularly in the kick’s initial stages, have been observed due to variations in stance positions [34,35]. Therefore, it is desirable to compare the differences between the FK and RK across various execution conditions.

Previous studies have examined the FK’s and RK’s dynamic forces and kinematic indicators under different conditions, including the target distance, target type, and participant’s experience. Therefore, this systematic review compares the FK and RK regarding maximum and impact forces, maximum velocity, maximum angular velocity, and execution time for different target distances, target types, and experience levels. Building upon previous studies, our hypothesis posits that the FK and RK will significantly differ in impact force and maximum velocity for different target types and experience levels.

This review compares the performance attributes of the FK and RK, explicitly focusing on practical use. There is an overview of the dynamic and kinematic differences among the different technical levels in connection with the different execution conditions of the FK and RK.

## 2. Materials and Methods

This article presents a systematic review that was conducted following the PRISMA (Preferred Reporting Items for Systematic Reviews and Meta-Analyses) recommendations [36,37]. The review protocol was prospectively registered online with PROSPERO (registration number CRD42022332589).

### 2.1. Literature Search

A comprehensive database search was conducted for the period from 1982 to 19 May 2022 using Web of Science, SportDiscus, and PubMed. The search strategy included specific keywords for a front kick (Mae-Geri and Apchagi) or a roundhouse kick (Mawashi and Dollyo), formulated according to each database’s requirements (see Appendix A). To ensure the inclusion of relevant articles, the reference lists of screened studies were also reviewed. The search was limited to articles written in English. All references were imported into Endnote X20 (Clarivate Analytics, Philadelphia, PA, USA), and duplicates were identified and removed.

### 2.2. Eligibility Criteria

To be included in the review, articles had to meet the following criteria: (i) they contained information on the front or roundhouse kick, (ii) they reported at least one of the dynamic forces or kinematic indicators (maximum force, impact force, maximum velocity, angular velocity, and execution time), (iii) they involved male participants, and (iv) they provided participants’ weight for calculating normalized outcomes when measuring dynamic forces. During the full manuscript screening, studies were excluded if (i) they reported the same results as another accepted study or (ii) they used simulation data instead of camera motion capture.

### 2.3. Study Selection

Two authors independently screened the titles and abstracts: the head combat instructor of the Czech Army (the first author) and the strength and biomechanics research expert (the last author). They decided which articles should be included in the full manuscript review. The first authors, together with the third and fourth authors, reviewed the full texts of the selected articles. Any discrepancies or disagreements among the authors were resolved through discussion and consensus. If needed, the second author made the final decision regarding article inclusion.

### 2.4. Data Collection Process

After selecting the relevant studies, the first author created an evidence table that included study demographic data, types of kicks and fighting activities, equipment used to measure dynamic and kinematic variables, and the variables themselves (see Appendix A). The third and fourth authors independently verified all the collected data.

### 2.5. Assessment of Methodological Quality and Risk of Bias

The first and last authors assessed the risk of bias in all articles in the systematic review. Any disagreements were resolved through discussion and consensus or by the decision of the second author. A scale was developed based on selected items from the STROBE (Strengthening the Reporting of Observational Studies in Epidemiology) Statement for observational studies [38], along with items created explicitly for this review. The assessment questions included the following: (1) Was the abstract an informative and balanced summary of the study? (2) Was the scientific background clearly explained? (3) Were the eligibility criteria, participant selection methods, and sources clearly stated? (4) Was the condition measured in a standardized and reliable manner for all participants? (5) Was the measurement of dynamic or kinematic indicators described in sufficient detail for replication? (6) Were any efforts to address potential bias described? (7) Were outcomes and conclusions clearly defined? The Cochrane Risk of Bias tool for the Generic dataset was used to display the risk of bias (see Appendix A), where the judgment was represented by symbols indicating a low risk of bias (+), some concerns (-), a high risk of bias (x), or no information (?).

### 2.6. Data Treatment

The means and standard deviations of dynamic forces (maximum force and impact force) and kinematic indicators (velocity, angular velocity, and execution time) were categorized based on the following criteria: experience level (elite, sub-elite, and novice), distance from the target, the height of the target, and stance position. The categorization rules were as follows:

Experience Level: Participants with a black belt or a combination of black and brown belts (the authors of some selected articles combined participants with brown and black belts into one group, and therefore, these groups were included in the elite group), international competitors, and head instructors of close combat in the army were classified into the elite group. Participants with martial arts degrees at the pupil level, different levels of martial arts degrees with a predominance of pupil degrees, national competitors, and soldiers with regular close combat training were classified into the sub-elite group. Participants without experience in fighting activities were classified into the novice group.

Distance from the Target: Studies that examined the effect of the target distance on select biomechanical parameters of kicks used an individual’s leg length to determine three distances (short, medium, and long) [12,13].

Height of the Target: The different heights of kicks were divided into two groups: the middle group (kick to the torso of the body) and the height group (kick to the head).

Stance Position: The stance position chosen was one leg in front and the second leg in the rear, and the kick was performed from the rear leg.

In studies that included an intervention program, data from the first session before the intervention were used for analysis. If the data in any of the studies were already normalized using the average weight of the participants, they were converted to the original units for descriptive statistics (we converted these data to the original data using the average weight and standard deviation of the probands that were stated in the study). To compare average values between groups (different levels of participants, different target types, etc.), the values of dynamic forces were normalized by the weight of the probands.

### 2.7. Statistical Analysis

Statistical analysis was performed using Statistica 14 (Tibco Software Inc., Palo Alto, CA, USA) and Microsoft Excel (Microsoft Corporation, Redmond, Washington, DC, USA). The Shapiro–Wilks test for data normality was used. The significance level alpha was set at ≤0.05. Forest plots and graphs were created using GraphPad Prism version 8.0. The mean and standard deviation of the maximum and impact forces of the front and roundhouse kicks were normalized by the participants’ weight and weighted by the number of participants. Similarly, the mean and standard deviation of the kick execution time, maximum velocity, and maximum angular velocity were also weighted by the number of participants. Forest plots were used to present the normalized weighted mean with a 95% confidence interval for dynamic variables and the weighted mean with a 95% confidence interval for kinematic variables. One-way ANOVA with Tukey’s post hoc test and Cohen’s *d* were used for comparison and effect size calculations. Levene’s test was employed to assess data equality of variance. In cases where two variables were compared, the *t*-test was utilized.

## 3. Results

A total of 619 records were retrieved through individual searches of Web of Science, SportDiscus, and PubMed, and 9 additional articles were identified from the reference lists. After screening titles and abstracts and removing 94 duplicates, 129 articles remained. The full texts of these studies were assessed, and after an objective assessment, 42 articles remained (1 article contained the FK and RK); 18 articles included the FK, and 25 included the RK (Figure 1). The records of the remaining 42 articles (martial arts = 2 articles; karate = 9 articles; Taekwondo = 23 articles; Musado = 5 articles; and 2 articles included karate Taekwondo and Muay Thai participants), which included a pooled sample of 113 elite men, 109 sub-elite men, and 46 novice men performing the FK and 238 elite men, 143 sub-elite men, 27 novice men performing the RK, were included in the systematic review and divided into individual categories according to dynamic and kinematic variables.

However, there were only enough data available to compare the levels of participants in terms of the impact force, the maximum velocity of the foot, knee, and hip, the execution time, and the maximum angular velocity of the knee and hip extension (Table 1 and Table 2).

### 3.1. Forest Plots of Front and Roundhouse Kicks

The graphical presentations of the pooled data are shown in forest plots separately for the FK and RK (Figure 2a–f and Figure 3a–d). The normalized weighted means with 95% CIs of the impact force are visually compared to show the differences among the levels of groups (novice, sub-elite, and elite) that executed both the FK and RK; in addition, these differences were observed at various distances from the solid target. Unfortunately, comparing the maximum forces of the FK was impossible due to the lack of studies with novice and sub-elite participants, and for the RK, the maximum force was not mentioned in any studies. Comparing the impact force between the FK and the RK shows that the impact force of the FK was higher than that of the RK (Figure 2d and Figure 3a).

Visually comparing the weighted mean with 95% CI of the maximum foot velocity executed from the middle distance into a solid target placed at a middle height showed slight differences between the sub-elite and elite groups for the FK (Figure 2a) and RK (Figure 3c). However, the lack of studies made comparing the maximum foot velocity between the sub-elite and novice groups impossible. When comparing the maximum foot velocity between the FK and RK, it is evident that the velocity of the RK was higher than that of the FK.

Comparing the maximum angular velocity of knee extension was possible only between sub-elite and elite groups that executed the kick from a middle distance into a solid target placed at a middle height (Figure 2e and Figure 3b). Comparing the maximum angular velocity of knee extension between the FK and RK revealed that the RK was faster than the FK.

Regarding the execution time, it could be calculated only for the elite group executing the FK into a solid target (Figure 2f) and at group levels for the execution of the first and second phases of the RK (from the start to hitting the target) at a middle distance into a solid target placed at a middle height (Figure 3d). In other cases, neither comparison was possible due to the lack of studies, or it was evident that there were no differences.

### 3.2. Comparing the Kicks’ Dynamic Forces and Kinematic Indicators

There were enough data to compare impact forces, the maximum velocity of the foot, knee, and hip, and the maximum angular velocity of knee extension, where the conditions for kick execution were set up at a middle distance from a solid target, which was placed at middle height.

### 3.3. Impact Force of the Kick

There were differences in the normalized weighted mean of the impact force among novice, sub-elite, and elite groups for the FK and RK, as shown in Figure 4a (*F*_5, 363_ = 84.15, *p* = 0.00001, μ^2^ = 0.54), where post hoc tests revealed that the impact force of the FK was higher than that of the RK between the novice, sub-elite, and elite groups (*p* < 0.01, *d* = 1.95; *p* < 0.01, *d* = 1.97; and *p* < 0.01, *d* = 1.71, respectively). Regarding differences among experience levels for the front kick, the elite group had a higher impact force than the sub-elite and novice groups (*p* < 0.01, *d* = 0.8 and *p* < 0.01, *d* = 1.87, respectively), and the sub-elite group had a higher impact force than the novice group (*p* < 0.01, *d* = 1.85). For the RK, the difference was between the elite and novice groups (*p* < 0.01, *d* = 1.3).

In terms of the distances from the target for the RK, there were differences in the normalized weighted mean of the impact force among the close, middle, and large distances for all experience levels (Figure 4b, *F*_8, 472_ = 15.517, *p* = 0.00001, μ^2^ = 0.21). Post hoc tests revealed that the elite group had a higher impact force than the sub-elite and novice groups at a close distance (*p* < 0.05, *d* = 0.66 and *p* < 0.01, *d* = 1.08, respectively), at a middle distance (*p* < 0.01, *d* = 0.57 and *p* < 0.01, *d* = 1.23, respectively), and at a large distance (*p* < 0.05, *d* = 0.73 and *p* < 0.01, *d* = 1.51, respectively). The sub-elite group had a higher impact force than the novice group at the middle and large distances (*p* < 0.01, *d* = 0.68 and *p* < 0.01, *d* = 0.66, respectively), and the novice group had a higher impact force at a close distance than at a large distance (*p* < 0.01, *d* = 0.4).

### 3.4. The Velocity of the Knee and Hip

There were differences in the weighted mean of the maximum knee velocity between the sub-elite and elite groups for the FK and the RK, as shown in Figure 4c (*F*_2, 210_ = 52.05, *p* = 0.0001, μ^2^ = 0.33), where post hoc tests revealed that the elite group had a higher maximum knee velocity during a roundhouse kick into a target than the sub-elite group and elite group (*p* < 0.01, *d* = 1.64 and *p* < 0.01, *d* = 1.04, respectively). However, including the novice group and sub-elite group in the test for the RK was impossible due to the lack of studies. Concerning the weighted mean of the maximum hip velocity, there were no differences between the elite and sub-elite groups for the FK and the elite group for the RK. The comparison of the knee and hip maximum velocities between the execution of the FK or RK into a solid target and the air was impossible due to the lack of studies.

### 3.5. The Velocity of the Foot

There were differences in the weighted means of the maximum foot velocity between the sub-elite and elite groups for the FK and RK, as shown in Figure 4d (*F*_3, 369_ = 439.7, *p* = 0.00001, μ^2^ = 0.78), where post hoc tests revealed that the elite group had a higher maximum foot velocity than the sub-elite group for the RK (*p* < 0.01, *d* = 1.24). The elite group performing the RK had a higher maximum foot velocity than the elite and sub-elite groups performing the FK (*p* < 0.01, *d* = 3.4; and *p* < 0.01, *d* = 4.21, respectively). The sub-elite group performing the RK had a higher maximum foot velocity than the elite and sub-elite groups performing the FK (*p* < 0.01, *d* = 2.52; and *p* < 0.01, *d* = 3.43, respectively). In addition, there were differences in the weighted means of the foot velocity for the FK, where the elite group had a higher maximum foot velocity than the sub-elite group (*p* < 0.01, *d* = 0.77).

Regarding the weighted mean of the maximum foot velocity during the execution of the kick into a solid target versus into the air, the maximum foot velocity of the FK into the air was higher than that of the kick into a solid target in the sub-elite group (*p* < 0.01, *d* = 2.42, Figure 5a), and the maximum foot velocity of the RK into a solid target was higher than that of the kick into the air in the elite group (*p* < 0.01, *d* = 1.36, Figure 5a).

Another possibility for the comparison was between the different styles of combat activities. The sub-elite group of Taekwondo participants had a higher maximum foot velocity (9.695 + 1.058 m/s, n = 32) than the sub-elite groups of karate (7.616 + 0.938 m/s, n = 42) and Musado (7.812 + 0.176 m/s, n = 53) participants when performing the FK (*p* < 0.001, *d* = 2.35; and *p* < 0.001, *d* = 1.93 respectively). The comparison of other groups was impossible due to the lack of studies.

### 3.6. Angular Velocity of the Knee and Hip

There were differences in the weighted mean maximum angular velocity of knee extension between the sub-elite and elite groups, as shown in Figure 5b (*F*_3, 122_ = 108.113, *p* = 0.00001, μ^2^ = 0.73), where the post hoc test revealed that the elite group had a higher maximum angular velocity than the sub-elite group for the RK (*p* < 0.01, *d* = 2.6) and the sub-elite and elite groups for the FK (*p* < 0.01, *d* = 3.39; *p* < 0.01, *d* = 3.26, respectively). The sub-elite group had a higher maximum angular velocity for the RK than the sub-elite and elite groups for the FK (*p* < 0.01, *d* = 1.08; *p* < 0.01, *d* = 1.16, respectively). However, there were not enough data for comparison with the novice group.

Regarding the maximum angular velocity of hip extension, there were differences between the FK and RK into a solid target placed at a middle height. The elite group performing the RK had a lower maximum angular velocity than the sub-elite and elite groups performing the FK, as shown in Figure 5c (*p* < 0.01, *d* = 4.14 and *p* < 0.01, *d* = 4.35, respectively). However, including the novice group in the comparison was impossible due to the lack of studies. There were no differences between the sub-elite and elite groups for the FK.

### 3.7. Execution Time

A comparison of the execution time among groups related to the participants’ experience and distance from the target was possible for the first and second phases together (from start to contact with the target) for the RK. There were differences in the weighted mean of the execution time among the close, medium, and large distances within the novice, sub-elite, and elite groups, as shown in Figure 5d (*F*_8, 505_ = 20.432, *p* = 0.00001, μ^2^ = 0.24). Post hoc tests revealed that the elite group had a shorter execution time than the sub-elite and novice groups with a large distance from the target (*p* < 0.05, *d* = 0.58 and *p* < 0.05, *d* = 0.52, respectively).

Regarding the differences within individual groups, the novice group had a longer execution time at a large distance than at a close distance (*p* < 0.01, *d* = 0.71), the sub-elite group had a longer execution time at a large distance than at middle and close distances (*p* < 0.01, *d* = 0.94 and *p* < 0.01, *d* = 1.2, respectively), and the elite group had a longer execution time at a large distance than at middle and close distances (*p* < 0.01, *d* = 0.76 and *p* < 0.01, *d* = 1.32, respectively). Comparing the execution times of FKs was impossible due to the lack of studies.

Regarding the comparison between target types, the first and second phases of the RK in the elite group were shorter for the execution of the kick into a target than into the air. However, the execution time was not significantly different between the phases of the kick (*p* = 0.210 and *p* = 0.213).

## 4. Discussion

This review identified significant differences between FKs and RKs in terms of the impact force and velocity for different target types. The main findings of this review support the hypothesis that the impact force of the FK is higher than that of the RK for all skill levels (novice, sub-elite, and elite) when kicking from a middle distance into a solid target placed at a middle height. Specifically, the impact force of the FK was 47% higher in the novice group, 92% higher in the sub-elite group, and 120% higher in the elite group compared to the RK. Additionally, the maximum foot velocity during the RK was higher than that during the FK for the sub-elite and elite groups, with a 44% increase in velocity for the sub-elite group and a 48% increase for the elite group.

Further analysis of the maximum foot velocity during the FK revealed interesting differences between the sub-elite and elite groups. The sub-elite group exhibited a 32% higher maximum foot velocity when kicking into the air compared to kicking into a solid target. In contrast, the elite group had a 14% higher maximum foot velocity when kicking into a solid target compared to kicking into the air when performing the RK.

When examining the maximum angular velocity of knee extension, the elite group showed a 37% higher maximum angular velocity compared to the sub-elite group when performing the RK. Moreover, the elite group exhibited a 65% higher maximum angular velocity of knee extension during the FK compared to the RK under the same conditions.

It is worth noting that the review identified several gaps in the existing literature. For example, there was a lack of studies comparing the maximum forces of FKs and RKs, particularly among novice and sub-elite participants. Additionally, the review highlighted the need for more research on the execution time and the maximum angular velocity of hip extension in both kicks. Addressing these gaps would provide a more comprehensive understanding of the biomechanical characteristics of the FK and RK.

### 4.1. The Impact Force of the Kicks

The impact force of kicks is examined in this section. The comparison was made for kicks executed at a middle distance from a target placed at a middle height for the FK and different distances for the RK. In the elite group, the normalized weighted mean of the impact force of the FK was 38% higher compared to the sub-elite group and 140% higher compared to the novice group (with only one selected study available for novices). Similarly, for the RK at a middle distance from a solid target, the normalized weighted mean of the impact force in the elite group was 20% higher than that in the sub-elite group and 60% higher than that in the novice group (Figure 4a). Moreover, for the RK, the impact force decreased as the distance from the solid target increased, and this was observed across all participant levels (close, middle, and large distances) (Figure 4b).

From a close distance, the novice, sub-elite, and elite groups exhibited impact forces that were 20%, 5%, and 20% higher, respectively, compared to a middle distance. Furthermore, they displayed impact forces that were 39%, 10%, and 16% higher, respectively, compared to a large distance. However, the only significant difference was observed between the close and large distances in the novice group. This finding is consistent with a study [14] in which no significant differences in impact force were found concerning the execution distance in expert competitors.

Contrary to the expectation of a higher impact force with a more extended time for action, the impact force decreased as the distance from the target increased. This finding is likely attributed to the stance position adopted by fighters when executing the RK from a long distance, which differs from the stance position when closer to the target. This supports the notion that when kicking from a long distance, the standing position is similar to a 90° stance position, limiting the RK’s effectiveness [35].

The impact force of the kick may be related to the isokinetic strength of the hip flexors and extensors and the knee angular velocity [2,51]. Therefore, fighters should focus on enhancing the impact force from a larger distance with the help of increasing their angular velocity during knee extension in the pre-contact phase [10].

When comparing the normalized weighted mean of the impact force between the FK and RK, it was observed that at a middle distance from the target, the FK had a higher impact force than the RK: 47% higher in the novice group, 92% higher in the sub-elite group, and 120% higher in the elite group.

While exploring data from the chosen articles, we also found that the impact force of the kick might be related to the isokinetic strength of the hip flexors and extensors and the knee angular velocity [51,60]. Therefore, we can recommend improving the impact force of kicks with the help of training involving functional exercises focused on the explosive strength of the lower limbs, where, in addition to traditional exercises, attention is also focused on exercises supporting the pre-contact and contact phases of the kick [2,61].

### 4.2. The Maximum Velocity of Kicks

The maximum velocity of kicks was examined in various studies, which presented different conditions for kick execution, including the target type, target height, distance from the target, stance position, and participant level. However, there were only enough studies to compare foot velocity during kicks executed into a solid target and into the air. The variations in conditions for kick execution across studies, such as target type, target height, and stance position, make it challenging to draw definitive conclusions.

When comparing the weighted mean of the maximum foot velocity between the FK and RK, it was found that the RK exhibited higher velocities than the FK at a middle distance from the target. Specifically, the RK was 44% higher in the sub-elite group and 48% higher in the elite group. This is consistent with the finding that circular kicks generate greater foot velocity at impact than linear kicks due to the rotational involvement of the segments in different planes [62]. Both kicks are characterized by proximo-distal movement; however, while the FK is primarily executed through hip movement in the sagittal plane and foot velocity is largely influenced by knee velocity [10,32,63], the RK achieves comparable hip velocity in the sagittal plane. However, when combined with the transversal and frontal planes, the RK attains higher velocity [25].

Furthermore, the weighted mean of the maximum foot velocity during the FK into a solid target was 11% higher in the elite group compared to the sub-elite group (Figure 4d). In this review, the maximum foot velocity range for the front kick varied from 7 to 9.98 m/s in the sub-elite group, from 8.2 to 10.32 m/s in the elite group, and 7.7 m/s in the novice group. Additionally, the weighted mean of the maximum foot velocity during the RK executed into a solid target at a middle distance was 13.3% higher in the elite group compared to the sub-elite group (Figure 4d). Differences between elite and sub-elite groups of participants were examined for both kicks. Several were discovered when exploring why elite participants achieve higher speed in the FK. For instance, they exhibited smaller angular ranges of leg movement and higher activation of the rectus femoris, vastus lateralis, biceps femoris, and gastrocnemius muscles [11]. Additionally, elite participants displayed lower variability in hip and knee joint movements [10,39]. In the case of the RK, the elite group reached higher angular velocities during hip and knee extension and ground reaction force [53,64,65]. Furthermore, better inter-joint coordination and muscle co-contraction were observed in the elite group [18,30,47].

In terms of comparing the execution of the FK into a solid target or the air, there is a study that found a higher foot velocity during the execution of the FK into a target than into the air among karate practitioners at the levels of brown and black belts [40]; however, our finding for the sub-elite group was consistent with a study [63] showing that the FK into the air demonstrated higher foot velocities than that into the target (Figure 5a). The weighted mean maximum foot velocity of the FK into the air in the sub-elite group was 31% higher compared to the foot velocity into the target. Conversely, for the RK in the elite group, the foot velocity during execution into a solid target was 14% higher than into the air (Figure 5a). This finding could be related to the protective action on the knee joint when performing a kick without impact, as it was found that while kicking into a target, the maximum activation of vastus lateralis was observed during knee flexion (first phase of the kick), and during a kick into the air, the peak Vastus Lateralis activity was reached during the knee extension phase, along with the antagonist activation of biceps femoris [65].

Regarding the maximum velocity of the knee in the FK and RK, the elite group exhibited a 6% higher maximum velocity of the knee during the FK compared to the sub-elite group. Furthermore, the elite group had a 13% and 20% higher maximum velocity of the knee during the RK compared to the sub-elite and novice groups during the FK, respectively (Figure 4c). Unfortunately, due to different conditions of kick execution, such as varying distances and heights of the target, further comparison of hip and knee velocities between the groups was impossible due to a lack of available data.

The results of this study reveal that sub-elite Taekwondo participants had a maximum foot velocity that was approximately 27% higher (9.695 m/s) compared to the sub-elite karate group (7.616 m/s) and about 24% higher compared to the sub-elite Musado group (7.812 m/s). These findings suggest significant differences in maximum foot velocity among the different combat styles, emphasizing the potential influence of training methodologies and techniques on achieving higher speeds. However, it is important to consider the study’s limitations, such as varying sample sizes and the focus on sub-elite participants, which may impact the generalizability of the results.

While exploring the data from selected articles, it was also found that to improve the maximum foot velocity in the execution of the FK, athletes need to increase the velocity of the knee traveling toward the target [7,32,63]. Investigating the optimal kicking techniques, such as the positioning of the supporting leg, the angle of hip and knee flexion, and the coordination of joint movements, can contribute to a deeper understanding of how athletes can generate higher foot velocities.

### 4.3. Angular Velocity of Kicks

The comparison of angular velocities between sub-elite and elite groups revealed significant differences only for the RK. The elite group had a 37% higher maximum angular velocity of knee extension compared to the sub-elite group when executing the RK toward a target at a middle height (Figure 5b). Additionally, a notable disparity was observed between the FK and RK, with the elite group demonstrating a 65% higher maximum angular velocity of knee extension in the RK compared to the FK. Conversely, within the FK, the elite group exhibited a 138% higher maximum angular velocity of hip extension compared to the RK (Figure 5c). When examining differences within the RK, it was observed that the elite group achieved a 39% higher maximum angular velocity of hip extension when executing kicks into the air compared to a target. Unfortunately, there were not enough studies to compare maximum angular velocities for knee and hip flexion in the RK and FK, specifically within the sub-elite group.

In the context of other studies, it was found that the maximum angular velocity of knee extension can be influenced by agonist and antagonist activities [33]. For example, the elite group exhibited clear antagonist activation of the biceps femoris during the extension phase of the FK, whereas such activation was not evident in the amateur group [30]. The elite group also achieved higher maximum angular velocities of the hip and knee compared to the amateur group. However, it should be noted that the antagonist activity in the biceps femoris toward the end of the analyzed time interval does not significantly affect the hip joint moment during the FK [31]. These findings partially explain the results showing an increase in angular velocity of knee extension after eight weeks of training focused on explosive lower limb strength without a significant increase in hip angular velocity [2]. Nevertheless, the primary movement of the hip is cited as crucial for the overall effectiveness of both the FK and RK [10,51]. Therefore, training to improve the angular velocity of the hip should also focus on functional exercises with an emphasis on core training [2,61].

Analyzing the contributions of individual muscles and their coordination within the lower limb complex can provide valuable insights into optimizing angular velocities and improving kick performance.

### 4.4. Execution Time of Kicks

The execution time of a kick is typically divided into three phases: the pre-phase, the attack phase, and the return phase [7]. The first two phases, known as the kicking time, were compared for the RK due to the selected articles in this systematic review. The findings revealed that the elite group had shorter kicking times compared to the sub-elite and novice groups across different distances. Specifically, the kicking time in the elite group was 5% shorter at a close distance, 10% shorter at a middle distance, and 12% shorter at a large distance compared to the sub-elite group. Similarly, compared to the novice group, the kicking time of the elite group was 7% shorter at close distance, 9% shorter at a middle distance, and 12% shorter at a large distance (Figure 5d). Notably, the greatest differences in kicking time were observed within each group at different distances, with significant reductions in kicking time from close to the middle and from close to large distances. The elite group exhibited the highest differences between close and middle distances.

The total time of the kick is primarily influenced by technique, as elite athletes demonstrate faster hip flexion, knee flexion, and extension compared to non-elite athletes [56,66]. Therefore, to achieve shorter kick times, focusing primarily on knee velocity in the FK [32,63] and the angular velocity of knee extension in the RK is advisable [18].

### 4.5. Limitations of the Study

Despite the valuable insights provided by the systematic review of kick biomechanics, there are certain limitations that should be acknowledged. The review may have excluded relevant studies due to specific inclusion criteria, such as language restrictions or different names for FKs or RKs. This could introduce a potential bias and limit the generalizability of the findings. The included studies may have used different methodologies, sample sizes, and participant characteristics, making it challenging to directly compare and synthesize the results. Therefore, the chosen criteria for classifying participants into the sub-elite and elite groups, where we classified brown belt participants into the elite group, can also be problematic, as some studies assign participants with brown and black belts to the elite group. The heterogeneity of the studies could impact the overall conclusions and limit the ability to draw firm conclusions. The lack of standardized measurement protocols and different measurement tools for measuring and analyzing kick biomechanics across studies may introduce variability in the data collection and analysis methods. This could affect the consistency and reliability of the results. The studies included in the review may have focused on specific populations, such as trained athletes or individuals of a specific age group. Therefore, the findings may not be applicable to other people, such as older adults or individuals with specific physical conditions. The review predominantly included cross-sectional studies, which provide a snapshot of kick biomechanics at a specific point in time. The review focused on front and roundhouse kicks, but there are various other types of kicks in combat sports and martial arts. The exclusion of different kick types limits the generalizability of the findings to a broader range of kicking techniques.

Addressing these limitations and conducting research with more rigorous methodologies, larger sample sizes, standardized protocols, and diverse populations will enhance the understanding of kick biomechanics and provide more robust evidence for training and performance optimization.

## 5. Conclusions

This systematic review revealed several important findings regarding the FK and RK. Firstly, the FK demonstrated a higher impact force compared to the RK across all experience groups, indicating its potential effectiveness in generating forceful strikes. On the other hand, the RK had a higher maximum foot velocity, suggesting its potential for swift and rapid execution. To improve the maximum foot velocity in the execution of the FK, fighters should increase the velocity of the knee traveling toward the target. The velocity of the knee in the first phase of the kick is mainly related to the technical execution of the kick. Therefore, to improve, fighters should primarily focus on the technical execution of the kick and functional exercises related to explosive strength.

Furthermore, differences were observed between the FK and RK when executed into a target or into the air. Specifically, the FK demonstrated a higher maximum velocity when performed in the air. Conversely, when performed into a target, the RK showed a higher maximum velocity.

In terms of maximum angular velocity, the RK displayed a higher maximum angular velocity of knee extension within the elite group. This highlights the importance of knee movement and extension in generating speed and power during the RK. However, it is worth noting that the FK exhibited a higher maximum angular velocity of hip extension compared to the RK within the same elite group, indicating the significance of hip movement in executing the FK with speed and efficiency.

To achieve a shorter kick time, it is advisable to focus mainly on the knee velocity in the FK and the angular velocity of the knee extension in the RK. Moreover, to improve hip movement and the hip’s angular velocity, fighters should focus on functional exercises with an emphasis on core training.

In summary, this systematic review provides valuable insights into the differences between the FK and RK in terms of impact force, maximum velocity, and maximum angular velocity. These findings contribute to a better understanding of the biomechanical aspects and a possible increase in efficiency when performing the FK and RK.

## Figures and Tables

**Figure 1 sports-11-00141-f001:**
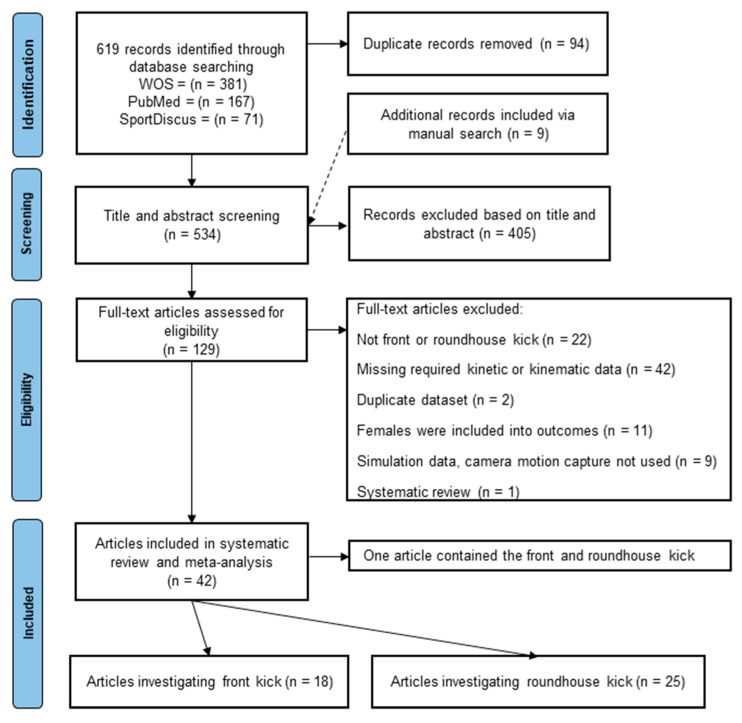
The flow chart of the systematic approach to the selection of articles relevant to front and roundhouse kicks.

**Figure 2 sports-11-00141-f002:**
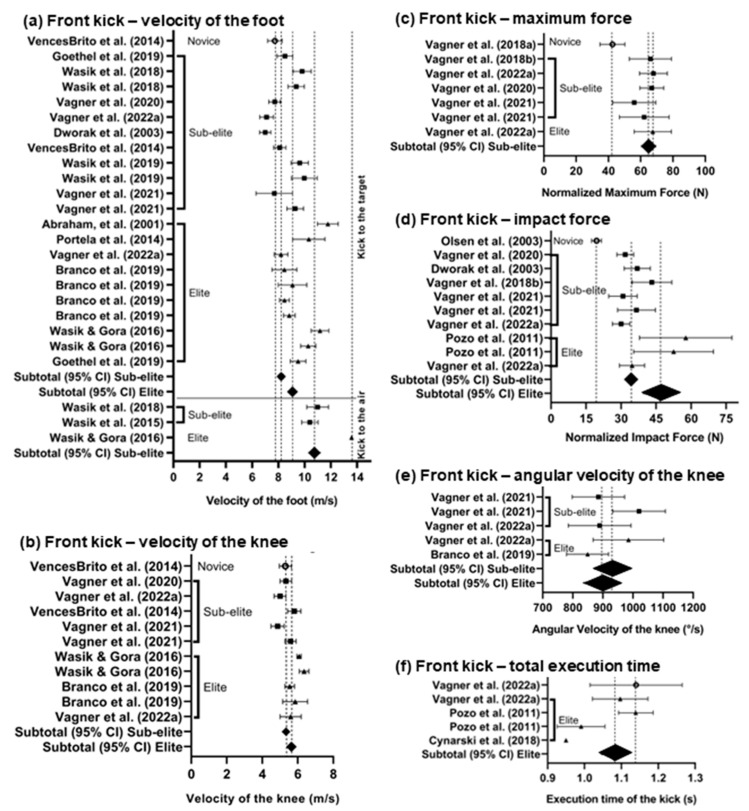
Forest plots presenting means and 95% confidence intervals: (**a**) front kick—maximum velocity of the foot; (**b**) front kick—maximum velocity of the knee; (**c**) front kick—maximum force; (**d**) front kick—impact force; (**e**) front kick maximum—angular velocity of the knee; (**f**) front kick—total execution time. Note: ○—novice; ■—sub-elite; ▲—elite; and ♦—weighted mean by the number of participants and their 95% CIs. References: [1,2,3,7,11,15,16,17,24,32,39,40,41,42,43,44,45,46].

**Figure 3 sports-11-00141-f003:**
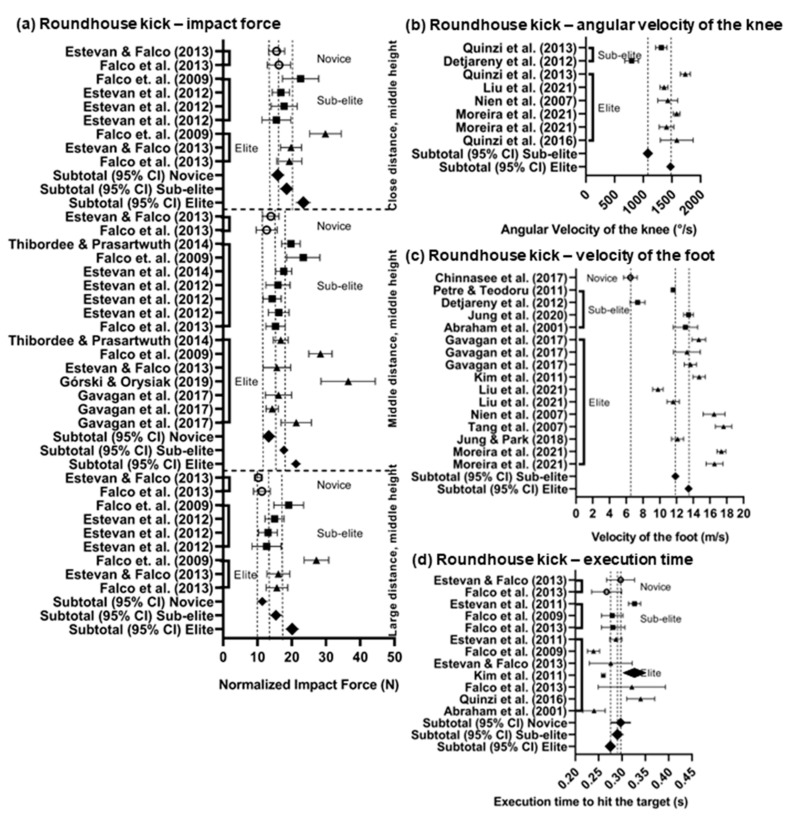
Forest plots presenting means and 95% confidence intervals: (**a**) roundhouse kick—impact force; (**b**) roundhouse kick—maximum angular velocity of the knee; (**c**) roundhouse kick—maximum velocity of the foot; (**d**) roundhouse kick—execution time. Note: ○—novice; ■—sub-elite; ▲—elite; and ♦—weighted mean by the number of participants and their 95% CIs. References: [12,13,14,18,22,23,28,34,35,44,47,48,49,50,51,52,53,54,55,56,57,58,59].

**Figure 4 sports-11-00141-f004:**
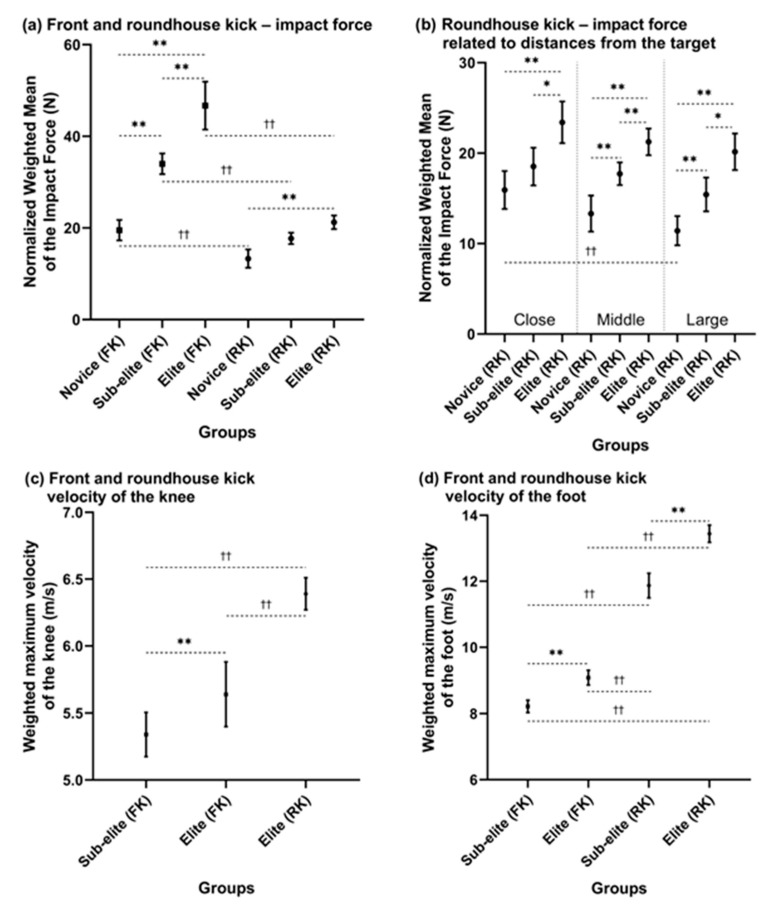
Differences among groups and distances from the target: (**a**) front and roundhouse kicks—impact force; (**b**) roundhouse kick—impact force related to distances from the target; (**c**) front and roundhouse kicks—velocity of the knee; (**d**) front and roundhouse kicks—velocity of the foot. Note: FK—front kick; RK—roundhouse kick; * significant differences among groups, *p* < 0.05; ** significant differences among groups, *p* < 0.01; ^††^ significant differences between the front and roundhouse kicks or distances from the target, *p* < 0.01. Vertical bars denote 95% CIs.

**Figure 5 sports-11-00141-f005:**
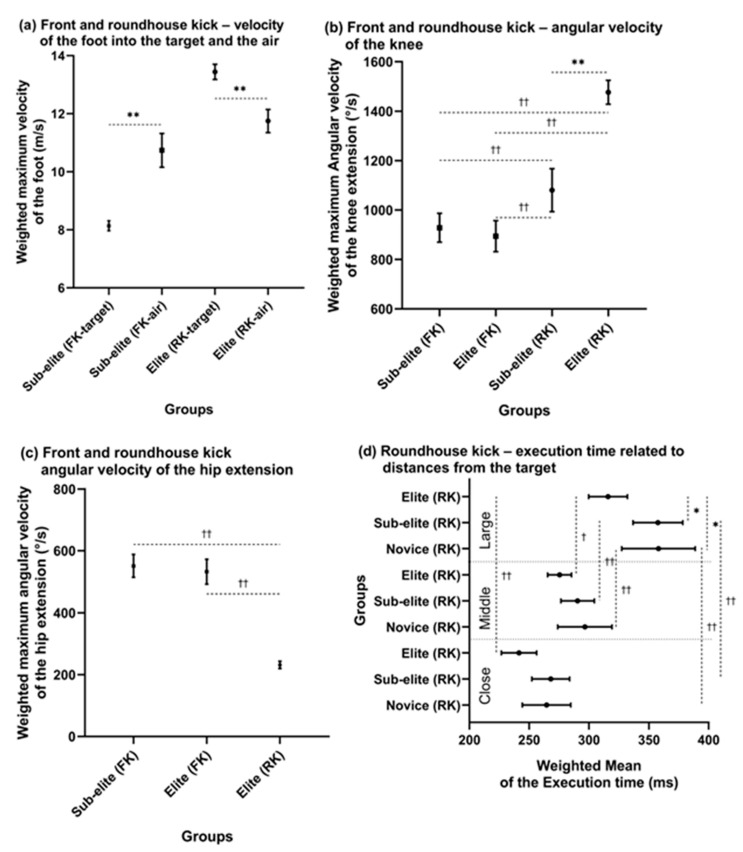
Differences among groups and distances from the target: (**a**) front and roundhouse kicks—velocity of the foot; (**b**) front and roundhouse kicks—angular velocity of knee extension; (**c**) front and roundhouse kicks—angular velocity of hip extension; (**d**) the roundhouse kick execution time related to distance from the target. Note: FK—front kick; RK—roundhouse kick; Close, Middle, and Large denote the distance from the target; * significant differences among groups, *p* < 0.05; ** significant differences among groups, *p* < 0.01; ^†^ significant differences between distances within one group, *p* < 0.05; ^††^ significant differences between front and roundhouse kicks or distances from the target, *p* < 0.01. Vertical or horizontal bars denote 95% CIs.

**Table 1 sports-11-00141-t001:** Performance of the front kick (mean and SD).

Front Kick	Novice	Sub-Elite	Elite	Novice	Sub-Elite	Elite
	Maximum Force (N)			Impact Force (N)		
Number of mean results	1	5	1	1	6	3
Number of participants	6	58	12	18	54	29
Participants’ weight (kg)	74.3	82.6	86.9	80.5	81.5	77.7
Front kick (M ± SD)	3157 ± 291	5261 ± 1497	5869 ± 1763	1570 ± 362	2846 ± 805	3696 ± 1621
	The velocity of the foot (m/s)—Target			The velocity of the foot (m/s)—Air		
Number of mean results	1	11	10	-	2	1
Number of participants	16	127	103	-	22	1
Front kick (M ± SD)	7.7 ± 1.2	8.56 ± 1.08	9.61 ± 1.05	-	10.74 ± 1.01	12.25 ± 0.18
	The velocity of the knee (m/s)—Target			The velocity of the knee (m/s)—Air		
Number of mean results	1	5	5	-	1	1
Number of participants	16	67	47	-	6	1
Front kick (M ± SD)	5.3 ± 0.7	5.32 ± 0.62	5.88 ± 0.6	-	5.06 ± 1.19	6.32 ± 0.06
	The velocity of the hip (m/s)—Target			The velocity of the hip (m/s)—Air		
Number of mean results	1	5	3	-	-	-
Number of participants	16	67	45	-	-	-
Front kick (M ± SD)	2.1 ± 0.3	2.4 ± 0.49	2.39 ± 0.57	-	-	-
	Angular velocity of the knee (°/s)—Target Extension			Angular velocity of the hip (°/s)—Target Extension		
Number of mean results	-	3	2	1	3	2
Number of participants	-	28	36	6	28	18
Front kick (M ± SD)	-	934 ± 145	917 ± 191	427 ± 8	556 ± 90	536 ± 72
	Total execution time (s)—Target			Total execution time (s)—Air		
Number of mean results	-	1	3	-	-	1
Number of participants	-	12	29	-	-	1
Front kick (M ± SD)	-	1.140 ± 0.221	1.076 ± 0.099	-	-	0.950

**Table 2 sports-11-00141-t002:** Performance of the roundhouse kick (mean and SD).

Roundhouse Kick	Novice	Sub-Elite	Elite	Novice	Sub-Elite	Elite
	Impact Force (N) close distance, middle height			Impact Force (N) middle distance, middle height		
Number of mean results	2	4	3	2	7	7
Number of participants	42	52	40	42	106	65
Participants’ weight (kg)	75.6	74.2	74.9	75.6	70.7	79.7
Roundhouse kick (mean, SD)	1204 ± 507	1333 ± 536	1705 ± 524	1007 ± 483	1227 ± 429	1656 ± 459
	Impact Force (N) large distance, middle height			Impact Force (N) close distance, large height		
Number of mean results	2	4	3	1	1	2
Number of participants	42	52	40	21	14	25
Participants’ weight (kg)	75.6	74.2	74.9	75.7	72	75
Roundhouse kick (mean, SD)	864 ± 392	1101 ± 487	1456 ± 466	1121 ± 368	1327 ± 167	1641 ± 219
	Impact Force (N) middle distance, large height			Impact Force (N) large distance, large height		
Number of mean results	1	1	2	1	1	2
Number of participants	21	14	25	21	14	25
Participants’ weight (kg)	75.7	72	75	75.7	72	75
Roundhouse kick (mean, SD)	1053 ± 356	1469 ± 135	1605 ± 267	864 ± 361	1203 ± 154	1521 ± 249
	The velocity of the foot (m/s)—Target middle distance, middle height			The velocity of the foot (m/s)—Air middle distance, middle height		
Number of mean results	1	4	11	-	-	2
Number of participants	15	33	110	-	-	30
Roundhouse kick (mean, SD)	6.49 ± 1.61	11.36 ± 1.11	14.34 ± 1.35	-	-	11.76 ± 1.07
	The velocity of the foot (m/s)—Target middle distance, large height			The velocity of the foot (m/s)—Air middle distance, large height		
Number of mean results	-	1	2	-	1	1
Number of participants	-	9	15	-	7	7
Roundhouse kick (mean, SD)	-	11.3 ± 0.8	13.79 ± 1.45	-	13.95 ± 4.16	16.29 ± 2.16
	The velocity of the knee (m/s)—Target middle distance, middle height			The velocity of the knee (m/s)—Air middle distance, middle height		
Number of mean results	-	-	8	-	-	2
Number of participants	-	-	99	-	-	30
Roundhouse kick (mean, SD)	-	-	6.73 ± 0.56	-	-	5.99 ± 0.54
	The velocity of the knee (m/s)—Target middle distance, large height			The velocity of the knee (m/s)—Air middle distance, large height		
Number of mean results	-	-	-	-	1	1
Number of participants	-	-	-	-	7	7
Roundhouse kick (mean, SD)	-	-	-	-	6.91 ± 1.78	7.96 ± 1.78
	The velocity of the hip (m/s)—Target middle distance, middle height			The velocity of the hip (m/s)—Air middle distance, middle height		
Number of mean results	-	-	7	-	-	2
Number of participants	-	-	93	-	-	30
Roundhouse kick (mean, SD)	-	-	2.36 ± 0.31	-	-	2.04 ± 0.11
	The velocity of the knee (m/s)—Target middle distance, large height			The velocity of the knee (m/s)—Air middle distance, large height		
Number of mean results	-	-	-	-	1	1
Number of participants	-	-	-	-	7	7
Roundhouse kick (mean, SD)	-	-	-	-	2.75 ± 0.77	2.83 ± 0.03
	Angular velocity of knee flexion (°/s) Target			Angular velocity of knee flexion (°/s) Air		
Number of mean results	1	1	5	-	2	2
Number of participants	15	6	45	-	13	13
Roundhouse kick (mean, SD)	959 ± 302	837 ± 140	893 ± 194	-	534 ± 126	956 ± 92
	Angular velocity of knee extension (°/s)—Target			Angular velocity of knee extension (°/s)—Air		
Number of mean results	-	2	6	-	2	2
Number of participants	-	11	51	-	13	13
Roundhouse kick (mean, SD)	-	1057 ± 130	1516 ± 181	-	1075 ± 186	1517 ± 107
	Angular velocity of hip flexion (°/s) Target			Angular velocity of hip flexion (°/s) Air		
Number of mean results	1	2	8	-	2	2
Number of participants	15	11	69	-	13	13
Roundhouse kick (mean, SD)	496 ± 130	276 ± 46	334 ± 110	-	334 ± 31	442 ± 107
	Angular velocity of hip extension (°/s) Target			Angular velocity of hip extension (°/s) Air		
Number of mean results	-	1	7	-	2	2
Number of participants	-	6	86	-	13	13
Roundhouse kick (mean, SD)	-	419 ± 99	297 ± 110	-	154 ± 71	325 ± 141
	Total execution time (s)—Target middle distance, middle height			Total execution time (s)—Air middle distance, middle height		
Number of mean results	1	2	1	-	-	1
Number of participants	11	43	12	-	-	7
Roundhouse kick (mean, SD)	1.06 ± 0.17	0.955 ± 0.115	0.840 ± 0.012	-	-	0.848 ± 0.110
	First-phase execution time (s)—Target middle distance, middle height			First-phase execution time (s)—Air middle distance, middle height		
Number of mean results	-	-	2	-	1	2
Number of participants	-	-	12	-	7	13
Roundhouse kick (mean, SD)	-	-	0.255 ± 0.143	-	0.320 ± 0.56	0.171 ± 0.201
	Second-phase execution time (s)—Target middle distance, middle height			Second-phase execution time (s)—Air middle distance, middle height		
Number of mean results	-	-	2	-	1	2
Number of participants	-	-	12	-	7	13
Roundhouse kick (mean, SD)	-	-	0.113 ± 0.077	-	0.200 ± 0.048	0.146 ± 0.019
	Execution time to hit the target (s) middle distance, middle height			Execution time to hit the target (s) large distance, middle height		
Number of mean results	2	3	7	2	3	4
Number of participants	42	65	87	42	65	53
Roundhouse kick (mean, SD)	0.297 ± 0.073	0.296 ± 0.048	0.280 ± 0.099	0.358 ± 0.047	0.368 ± 0.084	0.316 ± 0.059

## Data Availability

All data generated or analyzed during this study are included in this published article and Appendix A.

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
