# Peer review of "A Systematic Review of Dynamic Forces and Kinematic Indicators of Front and Roundhouse Kicks across Varied Conditions and Participant Experience"

_sports, 2023, doi:10.3390/sports11080141_

Round 1
Reviewer 1 Report
Dear Authors
You have written an interesting systematic review paper focusing on this s front kick and roundhouse kick including maximal and impact force, maximum velocity, maximum angular velocity, and execution time, at different target types and experience levels.
Overall the paper is nicely structured.
The abstract is well written.
The introduction presents the main rationale well.
Methods: Prisma guidelines were strictly followed.
Proper search engines were used and the keywords search strings are very well presented in the supplemental material.
Line 95 - how could 2 studies report the same results as another accepted study (if they are not from the same sample-research group)? Please elaborate. How many cases did you have like this? report
The inclusion criteria and the risk of bias assessment for the included papers add to the review quality.
Line 133 - experience
Why were brown belts considered an elite group? This is based on which criteria? back up your decision
In the inclusion criteria, you report that only male participants were included. However, in the first paragraph of the results section, you report female participants. What was it then?! Correct
Please report number of studies per striking sport/martial arts in the 1st paragrpah of results.
All figures are of poor quality. Please improve them.
I don't see any comparison/analysis of selected variables to the type of combat sport/martial art. You have karate, taekwando, musado and other groups. This would add imensly to the understanding of the diferences and how diferent sports affect the biomechanical characteristics. Please add this analsis.
In the conclusion you mention adds to the practical use - HOW? some more practical aplication or guidelines for coaches should be added to improve performance of these technical elements - here the comparison to specific sports would be more than usefull.
Overal a very solid paper that needs some more improvement from authors.
Kind regards
Minor editing of the English language is required.
Author Response
Dear Reviewer,
Thank you for your comments and recommendations for improving this article. We have incorporated your comments into the manuscript and provide an explanation of each recommendation below.
Line 95 - how could 2 studies report the same results as another accepted study (if they are not from the same sample-research group)? Please elaborate. How many cases did you have like this? Report
Answer:
Two articles were written from one measurement. Although their focus on the research questions differed, the results needed for our study were the same, and therefore we could not include both of them. These are the two discarded articles that are listed in Figure 1 as "Duplicate dataset":
1a) 19. Wąsik, J.; Mosler, D.; Ortenburger, D.; Góra, T.; Cholewa, J. Kinematic Effects of the Target on the Velocity of Taekwon-Do Roundhouse Kicks. J. Hum. Kinet. 2021, 80, 61–69.
vs
1b) Wąsik, J., Mosler, D., Ortenburger, D., & Góra, T. (2021). Stereophotogrammetry measurement of kinematic target effect as speed accuracy benchmark indicator for kicking performance in martial arts. Acta of bioengineering and biomechanics, 23(4). (Excluded from review)
2a) Jung, T.; Park, H. The Effects of Defensive Footwork on the Kinematics of Taekwondo Roundhouse Kicks. Eur. J. Hum. Mov. 2018, 40, 78–95.
vs
2b)Jung, T.; Park, H. The Effects of Back-Step Footwork on Taekwondo Roundhouse Kick for the Counterattack. Eur. J. Hum. Mov. 2020, 44, 129–145. (Excluded from review)
The inclusion criteria and the risk of bias assessment for the included papers add to the review quality.
Answer: risk of bias assessment is in Supplementary Figure 1 and inclusion criteria is in the paragraph "Assessment of methodological quality and risk of bias.
Line 133 – experience
This mistake was corrected.
Why were brown belts considered an elite group? This is based on which criteria? back up your decision
Answer:
The main reason for the sticking of black and brown belt participants into the elite group was that the authors of some chosen studies for this review include the black and brown belt participants in one sample. However, in martial arts, brown belts are often considered an elite group due to several criteria that support this classification. In many martial arts disciplines, students with brown belts are often entrusted with leadership roles, such as leading regular exercises or assisting instructors. While they have not yet achieved the highest rank (black belt), brown belts have usually undergone extensive training, honing their techniques and knowledge to a high degree. We added this also to the limitation of the study.
In the inclusion criteria, you report that only male participants were included. However, in the first paragraph of the results section, you report female participants. What was it then?! Correct
Answer: Thank you for this notice, we removed the mention of women from this paragraph.
Please report number of studies per striking sport/martial arts in the 1st paragrpah of results.
Answer:
We added studies for Karate, Taekwondo, Musado, and Muay Thai in the first paragraph of the results.
All figures are of poor quality. Please improve them.
Answer: Yes, we will edit the images in higher quality.
I don't see any comparison/analysis of selected variables to the type of combat sport/martial art. You have karate, taekwando, musado and other groups. This would add imensly to the understanding of the diferences and how diferent sports affect the biomechanical characteristics. Please add this analsis.
Answer: Thank you for this recommendation, we added comparing among Taekwondo/Karate/Musado for weighted maximum foot velocity within the front kick in the results and discussion. However, another comparison was impossible due to too many differences like experiences, distance from the target and height of the target, the type of the target, and flexion or extension at angular velocity.
In the conclusion you mention adds to the practical use - HOW? some more practical aplication or guidelines for coaches should be added to improve performance of these technical elements - here the comparison to specific sports would be more than usefull.
Answer: we added within the discussion the implications of the results related to other studies and in the part of conclusion several recommendations for practical improvement related to impact force, velocity, angular velocity, hip movement, execution time, and exercises.
Reviewer 2 Report
An interesting article that systematizes the current state of knowledge about the kinetics of front and turning kick. Comments:
line 30 and 33 Correct the brackets
line 71 This review "compares" ... Maybe better - This review "evaluate"...
line 76 Correct the brackets
line 116 Correct the brackets
line 133 Shouldn't the sentence begin "Experience level:" ?
line 153-166 How to check if the distribution of data is normal ? What type of t-test was used ?
line 407 Correct the brackets
line 421, 439, 441, 445, 448, 449, 469, 470 Correct the brackets
line 457 The correct way to quote
Author Response
Dear Reviewer,
Thank you for your comments and recommendations for improving this article. We have incorporated your comments into the manuscript and provide an explanation of each recommendation below.
line 30 and 33 Correct the brackets
Answer: We corrected this bracket
line 71 This review "compares" ... Maybe better - This review "evaluate"...
Answer: Thank you for the recommendation, however, We think that "compared" is a more appropriate word given the analysis used.
line 76 Correct the brackets
Answer: We corrected this bracket
line 116 Correct the brackets
Answer: We corrected this bracket
line 133 Shouldn't the sentence begin "Experience level:" ?
Answer: We corrected this mistake
line 153-166 How to check if the distribution of data is normal ? What type of t-test was used ?
Answer: For the assessment of the normal distribution of data, we used the Shapiro-Wilk test. We added this information to the Statistical analysis.
line 407 Correct the brackets
Answer: We corrected this bracket
line 421, 439, 441, 445, 448, 449, 469, 470 Correct the brackets
Answer: We corrected these brackets
line 457 The correct way to quote
Answer: We corrected
Reviewer 3 Report
Thank you for the opportunity to review. The article focuses on the analysis of dynamic forces and kinematic indicators in straight kicks and side kicks under different conditions and participant expertise levels. The authors aim to compare impact forces, maximum velocity, maximum angular velocity, and execution time of straight kicks and side kicks, considering various target types and levels of expertise. The study was conducted based on a systematic literature review, searching databases such as Web of Science, SportDiscus, and PubMed from January 1982 to May 2022, following the PRISMA guidelines. Normalized kick values were compared using one-way analysis of variance (ANOVA). Eighteen articles on straight kicks and twenty-five articles on side kicks that met the inclusion criteria were included. The article provides important insights into the differences in impact forces and kinematic indicators between straight kicks and side kicks, depending on participants' expertise levels. The results suggest that straight kicks generate greater impact force, while side kicks achieve higher foot velocity. Additionally, differences in angular velocities of knee and hip flexion were observed depending on the type of kick and expertise level. This article offers valuable information for trainers and combat sports athletes, who can utilize these findings to enhance kicking techniques. The methodology is consistently and clearly presented, and the discussion is comprehensive. Congratulations to the authors on their work. I believe it will contribute to the knowledge in the field.
Author Response
Dear Reviewer, Thank you for your evaluation of our article.
Reviewer 4 Report
First of all, I would like to thank the authors for the presented results of their investigation related to kicks. Also, I would like to thank the editor for the opportunity to review this manuscript.
The manuscript entitled “A Systematic Review of Dynamic Forces and Kinematic Indicators in the Front and Roundhouse Kicks Across Varied Conditions and Participant Experience” compares forces and kinematics characteristics between the two most common kicks in the existing literature. In my opinion, the authors present an interesting topic that falls within the aims and scope of the Sports journal, special issue Biomechanics and Sports Performances. Since the front and roundhouse kicks have been studied from various aspects, providing a systematic review of dynamic forces and kinematics indicators for putting new insight into the topic is always welcome.
Although the manuscript is well structured, some structural improvements should be made. First, defining the main idea more clearly is necessary, resulting in more concrete scientific and/or practical knowledge and recommendations. The way the manuscript is conceived, it is unclear what should be done in future research nor how these results could be applied in practice. Better reasoning is needed to consider different shot heights, expert levels, and on-target and point-blank performance.
The Abstract needs more information, especially in the last part, where the meaning of the results should be presented.
The Introduction section frames the study's focus and why it is needed. The Introduction should discuss why this is an important area that needs research. In addition, a stronger case of the importance of the study and how it adds to the literature is needed. In other words, a stronger answer to the "so what" question is necessary. Concise definitions are needed of the concepts mentioned in the Introduction section before indicating that they are an issue/problem in kicks realizing. This will allow readers to frame what is being presented and claimed.
The Materials and Methods section needs work. Study selection (Lines 97-104) and Data collection process (Lines 105-110) specified what specific author did that is redundant. In MDPI journals, there is a place at the end for that. Data treatment should be more explained. What was the rationale for introducing those experience levels? For example, the brown belt could be at the lower level compared to national-level competitors. Regarding the distance from the target, the review paper should be oriented more critically and innovative rather than just take over data from articles that deal with taekwondo athletes. I believe that we agree that distance from the target depends on the participants' anthropometry. Further, why were distances below 0.68 m, between 0.71 and 0.99 m, 1.01 and 1.34 m, and over 1.41 m excluded? Finally, how were normalized data converted to the original units?
The Discussion section needs work. Rather than repeating the results, a more detailed and in-depth discussion of the implications of the results is needed, including realistically what changes are recommended based on the current findings and why.
The revised paper should be proofread.
Author Response
Dear Reviewer,
Thank you for your comments and recommendations for improving this article. We have incorporated your comments into the manuscript and provide an explanation of each recommendation below.
The Abstract needs more information, especially in the last part, where the meaning of the results should be presented.
Answer: we added within the abstract a final sentence summarizing the results found. Unfortunately, more information due to the limit of a maximum of 200 words is not possible.
The Introduction section frames the study's focus and why it is needed. The Introduction should discuss why this is an important area that needs research. In addition, a stronger case of the importance of the study and how it adds to the literature is needed. In other words, a stronger answer to the "so what" question is necessary. Concise definitions are needed of the concepts mentioned in the Introduction section before indicating that they are an issue/problem in kicks realizing. This will allow readers to frame what is being presented and claimed.
Answer: we added two paragraphs in the introduction, in the first we define the need to investigate the kinematic and kinetic attributes when performing the front and roundhouse kick, and in the second added paragraph we present the findings for both kicks in relation to proximo distal movement and execution of kicks in different conditions.
The Materials and Methods section needs work. Study selection (Lines 97-104) and Data collection process (Lines 105-110) specified what specific author did that is redundant. In MDPI journals, there is a place at the end for that. Data treatment should be more explained. What was the rationale for introducing those experience levels? For example, the brown belt could be at the lower level compared to national-level competitors. Regarding the distance from the target, the review paper should be oriented more critically and innovative rather than just take over data from articles that deal with taekwondo athletes. I believe that we agree that distance from the target depends on the participants' anthropometry. Further, why were distances below 0.68 m, between 0.71 and 0.99 m, 1.01 and 1.34 m, and over 1.41 m excluded? Finally, how were normalized data converted to the original units?
Answer: We agree with you that at the end of the article, there is a contribution from individual authors and therefore these two paragraphs could be redundant. However, in these two paragraphs, the procedure and the course of the selection of articles are summarized and therefore we believe that it is appropriate to leave them in the article.
Answer: Regarding the brown belt participants, we made this decision as a result of the fact that the authors of some selected articles combine brown and black belt participants into one group, and therefore these groups were included in the elite group. We have added this explanation to the data treatment and study limitations section.
Answer: Concerning the distance from the target, thank you, yes you are right, these are studies by authors who dealt with this issue in taekwondo competitors and used the same methodology based on the anthropometry of the participants. We modified the text in this part based on interpretations of the related chosen articles.
Answer: Concerning the data that were normalized in the studies, there were only two studies we selected where the averages of the selected data were normalized by the weight of the participants. We converted these data to the original data using the average weight and standard deviation of the probands that were stated in the individual studies. For the other chosen studies, original data were always provided and therefore it was not necessary to perform this procedure. We added this explanation in the data treatment.
The Discussion section needs work. Rather than repeating the results, a more detailed and in-depth discussion of the implications of the results is needed, including realistically what changes are recommended based on the current findings and why.
Answer: we added more information to individual paragraphs based on previous findings from studies dealing with this issue in the part of discussion, and practical recommendations in the part of conclusion.
Round 2
Reviewer 1 Report
Dear Authors,
Thank you for addressing all of my questions and suggestions. The quality of your manuscript improved. Therefore, I recommend acceptance.
Kind regards
Minor editing of the English language required
Reviewer 4 Report
Dear Authors,
Thank you for your time and effort to improve the manuscript. Now, it sounds much better.
Best regards.